# The Prototype of a Fast Vertical Ionosonde Based on Modern Software-Defined Radio Devices

Alexei V. Shindin [1,2,*], Sergey P. Moiseev [1,2], Fedor I. Vybornov [1,3], Kseniya K. Grechneva [2], Viktoriya A. Pavlova [1] and Vladimir R. Khashev [2]

1 Radiophysics Research Institute, Lobachevsky State University of Nizhni Novgorod, Bolshaya Pecherskaya St. 25/12a, 603950 Nizhni Novgorod, Russia; moiseev@rf.unn.ru (S.P.M.); vybornov@nirfi.unn.ru (F.I.V.); pavlova@nirfi.unn.ru (V.A.P.)

2 Department of Radio Propagation and Radio Astronomy, Radiophysics Faculty, Lobachevsky State University of Nizhni Novgorod, Gagarin Av. 23, 603950 Nizhni Novgorod, Russia; grekhneva@rf.unn.ru (K.K.G.); khashev@rf.unn.ru (V.R.K.)

3 Department of Physics, Volga State University of Water Transport, Nizhni Novgorod, 5 Nesterova Str., 603950 Nizhni Novgorod, Russia

* Correspondence: shindin@rf.unn.ru

**Abstract:** The description and test results of the prototype of a fast ionosonde for the vertical sounding of the ionosphere, which makes it possible to record ionograms once a second, are presented. Such a high rate of registration of ionograms is required to study the fast processes of redistribution of electron concentration during heating experiments, for registration of fast quasiperiodic and moving ionospheric disturbances in the F, E, and Es layers. The key feature of the presented development is the usage of publicly available radio-electronic components. This provided a significant reduction in the cost of creating the prototype. In the current version, the prototype is based on the software-defined radio (SDR) devices Red Pitaya SDRlab 122-16 and LimeSDR. The test results showed that the quality of the ionograms recorded using the prototype is not worse than the quality of ionograms recorded using the professional CADI ionosonde. The low cost of the components allows providing multi-position registration of ionograms for determination the dynamics of natural and artificial ionospheric disturbances in 3D region of space at a lower expenses rate, as well as to create a network of ionospheric observation points with an increased number of ionosondes.

**Keywords:** ionosphere; vertical pulse sounding; ionosonde; ionogram; software-defined radio

## 1. Introduction

Over the past 20 years, software-defined radio (SDR) devices have expanded its applicability from professional to amateur radio (see, for example, [1]). This was facilitated by the following factors: cheaper hardware components and the development of field-programmable gate array (FPGA) technology. The latter factor allowed extreme simplification of the structure of SDR devices for the RF range, factually excluding application-specific integrated circuits (ASIC) chips from it. Moreover, a substantial role was played by the fact that development environments for FPGAs from major vendors became free for individual use. Modern ADC/DAC with a sampling rate of more than 80 MHz allow the entire RF range (3–30 MHz) to be transmitted to the FPGA for processing. All of the above makes such devices extremely convenient for monitoring the passage of HF radio waves through the ionosphere and, in particular, for the implementation of the chirp ionosonde [2,3]. To monitor the current ionospheric situation and structure of the ionosphere, the reconstruction of the electron density profile, the most common technique is the vertical sounding of the ionosphere with short coded pulses with a filling frequency varying within 1–20 MHz. This paper presents the description and test results of the vertical sounding ionosonde prototype based on currently available SDR devices. The developed prototype can be used by scientific groups to create their own devices for monitoring ionospheric conditions.

## 2. Materials and Methods

### 2.1. Hardware Part

The vertical sounding ionosonde is an HF radar station (for more details, see [4]). During operation, the ionosonde emits short radio pulses with a filling frequency ranging from 1 to 20 MHz. As a rule, various types of manipulations are applied to the emitted pulses (usually amplitude or phase). The pulses reflected from the ionosphere are recorded and processed using the receiving part. As a result of signal processing a height-frequency characteristic is obtained, so that it is possible to restore the electron concentration profile in the altitude range of 80–700 km.

While developing the ionosonde prototype for vertical sounding, we focused on the technical characteristics of the CADI ionosonde [5] at our disposal: the radiation power is 600 W, the type of pulse encoding is phase shift keying with a 13-bit Barker code with a duration of one bit of 40 µs, the pulse repetition rate is from 25 µs, the range of probing frequencies is from 1 to 20 MHz with a step of 50 kHz in standard mode. In [6], you can find examples of ionograms obtained using the CADI ionosonde. It should be noted that in order to increase the signal-to-noise ratio when registering a signal reflected from the ionosphere, our CADI ionosonde uses averaging over 4 pulses emitted with the same filling frequency. In general, this requires approximately 40 s to emit all the pulses, and then one minute to process the signal and record the ionogram data into the file. The height resolution of ionograms obtained using the CADI ionosonde is 3 km.

The block diagram of the developed prototype of the fast vertical-sounding ionosonde is shown in Figure 1. In the current version, the transmitting and receiving parts of the ionosonde are separate devices. The transmitting part includes (1) DIY HF linear amplifier on LDMOS transistors [7] with a maximum power of 600 W with an operating range of 1.8–72 MHz; (2) 5 W pre-amplifier with an operating range of 100 kHz–40 MHz; (3) programmable attenuator with an operating range of up to 6 GHz and a maximum attenuation of 30 dB; (4) SDR device Red Pitaya SDRlab 122-16 [8]. Figure 2 shows a photo of the transmitting part of the ionosonde mockup without housing on a laboratory bench.

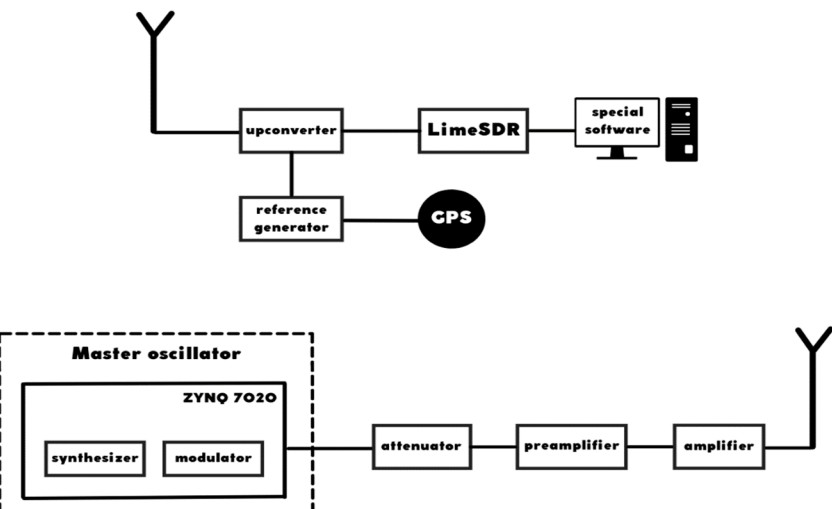

**Figure 1.** Block diagram of the transmitting (**bottom**) and receiving (**top**) parts of the ionosonde.

The Red Pitaya SDRlab 122-16 is the development board based on a Xilinx Zynq 7020 system that combines FPGAs and a general-purpose dual-core ARM processor. The board is equipped with two-channel 14-bit DAC and 16-bit ADC with a sampling rate of 122.88 MS/s, as well as Ethernet and USB2.0 interfaces for data transfer, communication with other devices (PC), and for connecting additional devices to the board (e.g., WI-FI dongle). This board is used as a master oscillator (probe pulse generator). We plan to use this device as a basis for the receiving part of the ionosonde in the future.

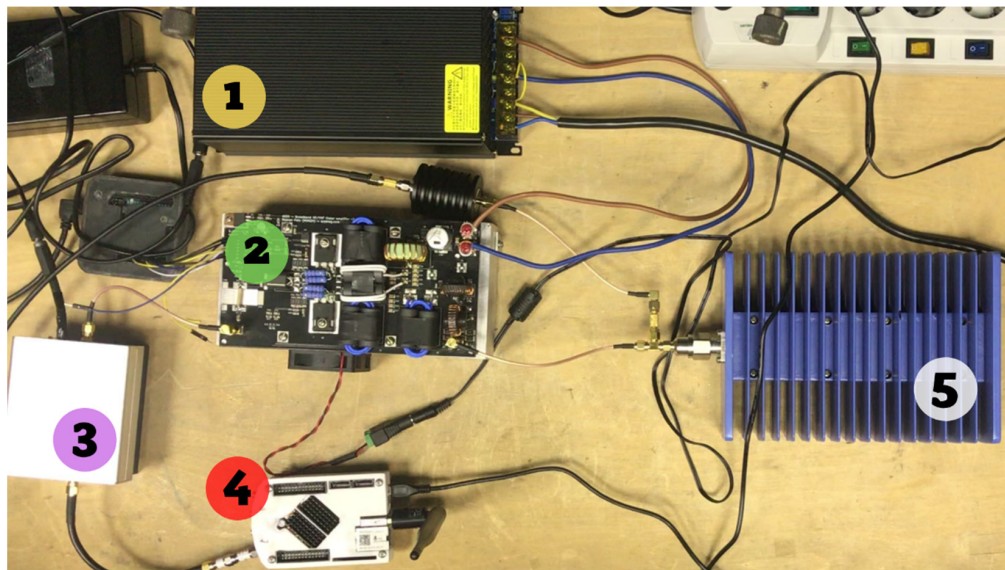

**Figure 2.** The photo of the assembled model of the transmitting part of the ionosonde without a housing. The numbers indicate 1—amplifier power supply, 2—linear amplifier A600, 3—5 W preamplifier, 4—SDRlab 122-16 in a 3d-printed case as a master oscillator, 5—matched load.

The receiving part of the developed layout (see Figure 3) is based on a two-channel SDR device LimeSDR [9] with a declared operating frequency range from 100 kHz to 3.8 GHz, which is capable of recording a signal in the 61.44 MHz bands. To be able to register the HF signal in the 10 MHz bands modified upconverters were used filtering the HF signal and moving it to the frequency range 120–130 MHz, where the LimeSDR can work more efficiently. Since the operation of the ionosonde assumes precise frequency matching between the transmitting and receiving parts, we used the Leo Bodnar precision GPS reference clock to generate the reference signal for the LimeSDR and upconverters. Two channels of the device allow, in presence of appropriate antennas and a polarizer, to register two polarizations of the reflected signal (O and X modes). A low noise amplifier (+20 dB) was used to amplify the received signal. To record a signal in the 10 MHz bands, the LimeSDR was connected to a PC or laptop via the USB3.0 interface.

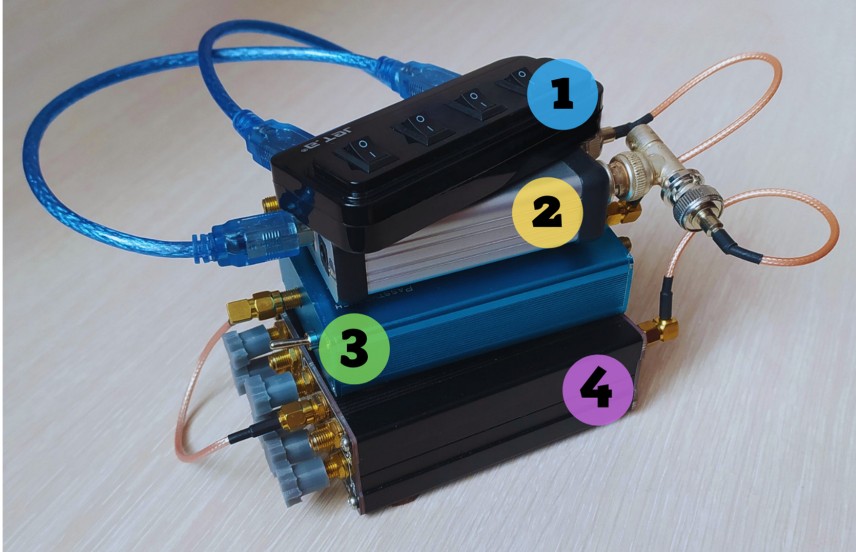

**Figure 3.** Photo of the assembled model of the ionosonde receiving part without a control PC. The numbers indicate: 1—USB hub, 2—precision GPS reference clock, 3—upconverter, 4—LimeSDR.

### 2.2. Software Part

The SDR concept assumes that the functionality and even the purpose of the device is determined by software components that can be easily changed or upgraded. In our case, the function and operation speed of the vertical sounding ionosonde is provided by the development of firmware for the SDRlab 122-16 FPGA board and software for signal processing and obtaining ionograms.

To achieve the ionogram recording time of the order of 1 s, we: (1) used a pulse repetition period of 5 ms instead of 25 ms (as in the CADI ionosonde); (2) abandoned the averaging of signals over four pulses at one frequency; (3) reduced the frequency range of sounding to 10 MHz; (4) developed software that allows recording ionograms in real-time, i.e., with a delay less than the time required for the emission of all probing pulses. All these measures ensured a sounding time of 0.9 s.

The functional block diagram of the master oscillator firmware on the SDRlab 122-16 board is shown in Figure 4. The master oscillator firmware was implemented by means of the Xilinx Vivado development environment in the Verilog hardware description language using embedded Xilinx IP cores. The project consists of separate modules that have a functional connection with one another: the input clock frequency of 122.88 MHz, coming from the crystal oscillator of the Red Pitaya board, is fed to the input of the clk_wizard module, where it is converted to the frequency of 100 MHz. The clock pulse is fed to the input of the clk_divider module and is divided into two clock frequencies: 25 kHz is the symbol frequency (1 symbol corresponds to 1 bit of a 13-bit Barker code with a duration of 40 μs) and 200 Hz is the frequency of the probing pulses (200 Hz corresponds 5 ms sounding pulse period). These two frequencies provide AM and FM control. In the FM_Modulator module, the phase increment of the harmonic signal changes every 5 ms with a frequency step of 50 kHz (to ensure the ionogram recording time—0.9 s), after which the increment is fed to the input of the DDS digital computational synthesizer module, where the cosine values are generated. The received harmonic signal enters the amplitude modulation module and the already modulated pulse is fed to the input of the digital-to-analog converter control module. Phase shift keying was implemented with a slight upgrade of the AMK firmware, taking advantage of the fact that changing the phase by $\pi$ is equivalent to multiplying the signal by $-1$. Note that the selected firmware parameters implement the radiation scheme of the CADI ionosonde, which we were guided by.

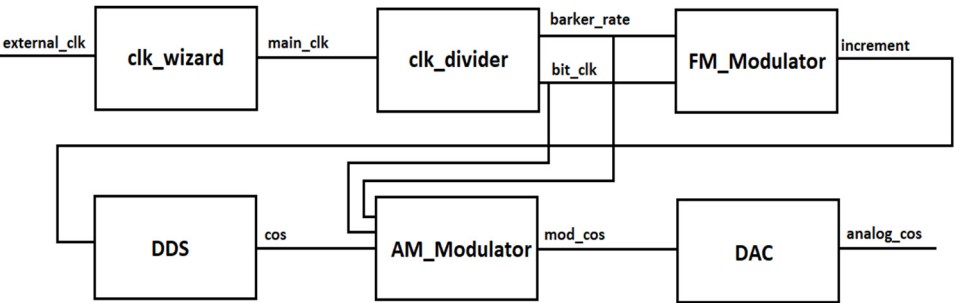

**Figure 4.** Functional block diagram of the master oscillator firmware on the SDRlab 122-16 board.

The software for recording ionograms was created on the basis of the GNU radio framework [10]. In addition, a trigger module was developed to compensate for lacking synchronization between the transmitting and receiving parts of the model at this stage, as well as possible missing samples. The used flow graph of the GNU Radio Companion tool is shown in Figure 5. An additional program written in python using the libraries numpy [11], scipy [12], and matplotlib [13] was used for autocorrelation analysis and ionograms. The developed software makes it possible to register ionograms with a height resolution of 1.5 km. The computing power of a laptop equipped with a quad-core central processor is sufficient to obtain an ionogram in less than 1 s, which in fact provides real-time

monitoring of the ionosphere. A separate application (for example, OBS Studio [14]) can be used to quickly publish ionograms on the Internet video services such as YouTube.

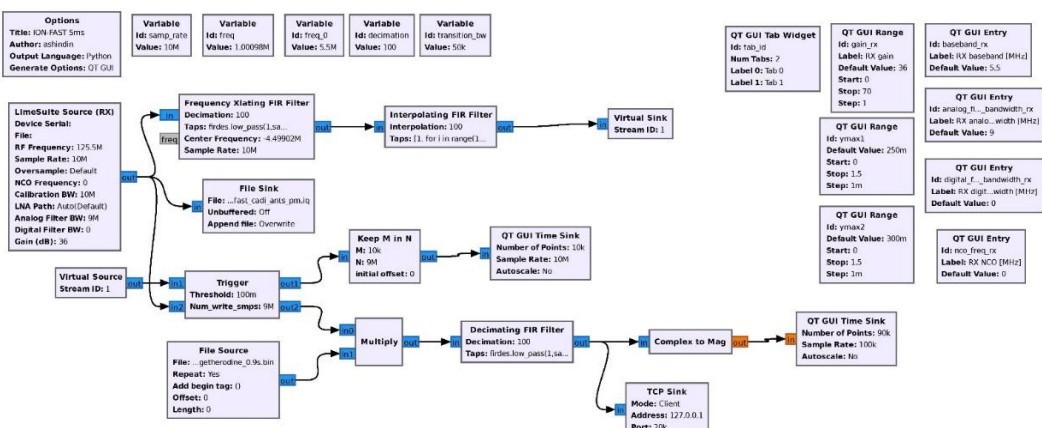

**Figure 5.** GNU Radio Companion flowgraph used in ionosonde's receiving part.

## 3. Test Results

Several series of tests were carried out for the developed prototype of the fast ionosonde. The first series of experiments included continuous operation of the ionosonde for 24 h at the maximum ionogram recording rate. Tests have shown that the thermal regime and performance of the components are not disturbed under prolonged loads. We have to mention that the computing capabilities of the hardware used are sufficient to obtain ionograms and their automatic publication on the Internet. The second series of experiments consisted in using several types of receiving antennas as part of the ionosonde. Among them: (1) a large diagnostic transmitting/receiving antenna (in-phase horizontal antenna array 126 × 126 m in size, suspended on 12 masts 16 m high; each of the two linear polarizations has 12 emitters; each emitter consists of three dipoles of different length connected in parallel, due to which the antenna has three resonant frequencies of 2.95, 4.6 and 5.7 MHz), that used to register artificial radio emission of the ionosphere, multifrequency Doppler sounding, diagnostics of the lower layers of the ionosphere and mesosphere, etc. (see Figure 6); standard receiving (two broadband crossed dipoles on four 12 m masts with an operating range of 2–10 MHz) antenna of the CADI ionosonde available at the Vasilsursk experimental base (see Figure 8); the receiving-transmitting Delta type antenna of the chirp ionosonde on a 15 m mast with the northern directional pattern and with an effective operating range of 4–15 MHz. Note that in all the tests, the standard transmitting antenna of the CADI ionosonde was used as the transmitting antenna (vertical delta antenna on a 40 m mast with an operating range of 2–30 MHz). All transceiver equipment was located at a distance of no more than 1 km from each other. It is planned to use the T2FD antenna for the mobile version of the ionosonde. The developed ionosonde prototype can be used together with any broadband HF antenna operating in the range of 1.8–10 MHz and vertical antenna pattern.

The receiving part of the developed ionosonde prototype in various test sessions included either an Acer Predator Helios 500 laptop with the Manjaro Linux 21 operating system installed or an ECS Liva SF110-A320 platform with an AMD Ryzen 5 PRO 2400GE quad-core processor.

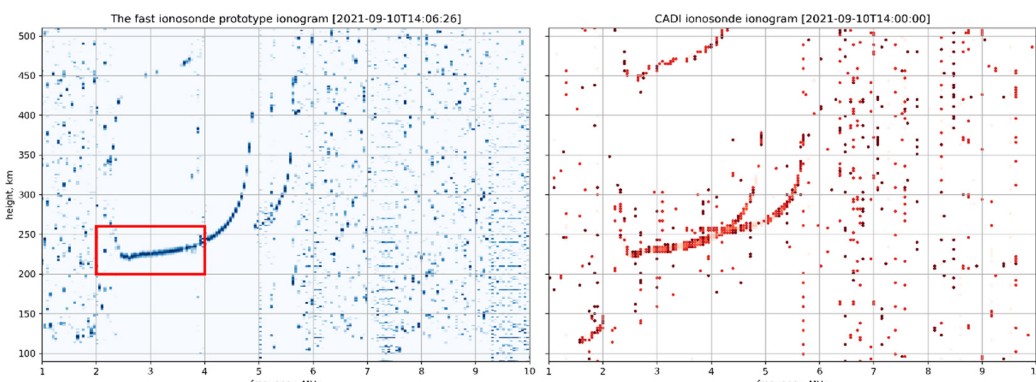

**Figure 6.** (**Left panel**): an example of an ionogram recorded with the developed ionosonde prototype using a diagnostic antenna (O mode output). Red square corresponds to the ionograms's subarray which presented in detail in Figure 7. (**Right panel**): an example of an ionogram registered with a CADI ionosonde. Ionogram's registration times are shown in UTC at the top of the panels.

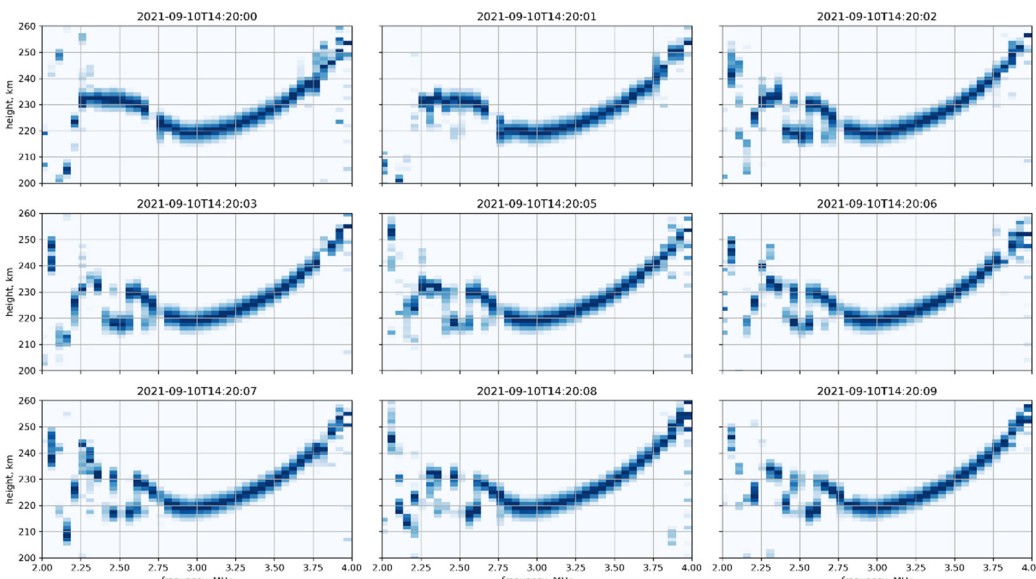

**Figure 7.** An example of 9 consecutive ionograms recorded at a rate of 1 ionogram per second, illustrating the effect of a rapid decrease in the effective reflection height in the F region of the ionosphere.

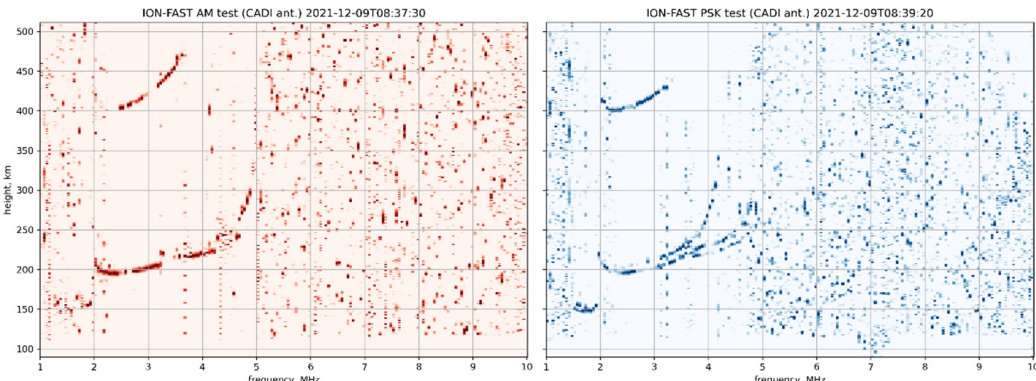

**Figure 8.** Examples of ionograms recorded using the developed prototype ionosonde using CADI ionosonde antennas. (**Left panel**): Barker code amplitude manipulation. (**Right panel**): Barker code phase shift keying.

Comparative analysis of the obtained ionograms showed that the diagnostic antenna, due to its good parameters, provides ionograms with the highest signal-to-noise ratio. Figure 7 shows 9 consecutive ionograms registered by the developed ionosonde prototype using a diagnostic antenna with a time resolution of 1 s. The figure shows the process of an abrupt change in the reflection height by a value of about 15 km in the ionosphere F layer, which could not be detected at the standard ionogram registration rate. Video files demonstrating the operation of the developed ionosonde prototype can be found in the Supplementary Materials. The standard receiving antenna of the CADI ionosonde required the use of a low-noise amplifier (we used 20 dB 1–30 MHz amplifier) to obtain ionograms of comparable quality. When using an inclined antenna of the chirp ionosonde, it was impossible to obtain ionograms with distinguishable traces of ionospheric layers at this stage of testing. In this case, oblique-sounding ionograms for this antenna are successfully recorded. In the third series of tests, two modes of operation of the ionosonde were compared: with amplitude and phase manipulations of the probe pulses. In both cases, the encoding was carried out with a 13-bit Barker code. Tests have shown that phase shift keying ionograms have a significantly higher signal-to-noise ratio. Apparently, this is due to the higher average sounding power in this mode.

## 4. Discussion

As can be seen from Figures 6–8, the developed prototype of the vertical ionosonde is capable of registering ionograms comparable in quality to ionograms obtained using the CADI ionosonde. At the same time, the developed prototype uses practically the same temporal radiation pattern and the same transmitter power. The difference in the representation of ionograms by the two instruments is due to many factors. Among the main ones are different characteristics of the receiving parts (the CADI ionosonde has an 8-bit ADC in the receiver) and different approaches to obtaining ionograms (the CADI ionosonde uses fast Fourier transform). The ionograms obtained by the developed ionosonde are quite suitable for further analysis (scaling) in order to reconstruct the electron density profile. The high rate of ionogram registration makes it possible to determine the parameters of fast movements in the ionosphere.

The fast ionosonde can be used to study fast natural variations of electron density profiles in the F-layer of the ionosphere. For example, the use of the fast ionosonde with an ionogram recording time of 2 s allowed detecting local disturbances moving vertically at speeds up to 50 m/s [15]. The horizontal stratification found in this case is capable of performing cyclic vertical movements with an amplitude of up to 5 km and a period of about 90 s.

It is known that traveling ionospheric disturbances (TIDs) with a characteristic spatial scale of up to 100 km are often observed at mid-latitudes in the daytime. Ionosondes operating according to the standard 15-min ionosphere sounding program record them on single ionograms (see, for example, [16]). The rare territorial location of ionosondes does not allow to determine TIDs parameters unambiguously. The relevance of the problem is explained by the difficulty of predicting TIDs and the strong influence on the HF communication channels. The use of linear frequency modulation (LFM) ionosondes and a one-minute sounding cycle made it possible to determine the spatial and dynamic characteristics of the TID from several (usually 5–15) ionograms [17]. As a rule, in this case, a weekly inclined mode of sounding the ionosphere is used by a system of synchronously operating chirp stations [18,19]. Reducing the sounding time to a few seconds would enable tracing the dynamics of the TID propagation processes in detail, especially if several closely located (at a distance of 50–100 km) automatic synchronously operating ionospheric stations were used. The use of several low-cost fast vertical-sounding ionosondes makes it possible to create such promising automatic systems for recording the TID parameters.

The fast vertical-sounding ionosonde can be useful in the development of studies of the processes occurring in the E and Es layers of the ionosphere. It is known that the processes occurring in these layers are characterized by fast dynamics [20].

Experiments on modifying the Earth's ionosphere with powerful short-wavelength radiation [21] have shown the need to develop and use fast vertical-sounding ionosonde. Firstly, the ionosonde in such experiments is used for diagnostic purposes (determination of the height of reflection of a powerful wave) and control of the operation of the heating facility (selection of the frequency of powerful transmitters). As a source of impulse noise, the ionosonde should work periodically but for a short time without interfering with the operation of the diagnostic equipment. Secondly, a high-speed ionosonde is required for research purposes. The interaction of high-power HF radiation of ordinary polarization is accompanied by the excitation of artificial ionospheric turbulence (AIT). There are several stages of its development with characteristic times (the development of striction parametric instability—5–20 ms, the stage of restoration of the level of the reflected signal of the pump wave—0.5–3 s, anomalous attenuation—0.5–10 s, the development of self-focusing non-stability—10–30 s) [21]. Typically, AIT surveys are conducted at fixed frequencies using a probe wave transmitter. The use of a new fast vertical-sounding ionosonde can be useful in investigating the properties of AIT.

## 5. Conclusions

In this paper, we showed that it is possible now to assemble the prototype of vertical sounding ionosonde using publicly available radio-electronic components with the total cost of approximately 1600 EUR (the prices are of December 2021), which has an ionogram recording rate of 1 ionogram per second.

This cost does not contain the cost of a PC for recording ionograms, as well as transmitting and receiving antennas. Moreover, the transmitting part of the prototype costs 1050 EUR. We plan to implement the receiving part on the SDRlab 122-16 board in the future. It will reduce the total cost by 550 EUR. Unfortunately, the CADI ionosonde is currently not available for order. However, at the time of purchase to equip the Vasilsursk base in 2007, it cost about 50,000 USD.

As we can see from Figure 6 (as well as from the demonstrations in the paper Supplementary File and data set), the developed ionosonde prototype allows obtaining ionograms at an unprecedented speed, comparable in quality to CADI ionosonde ionograms, at a significantly lower cost. The authors are not aware of any cases of demonstration of continuous long-term operation of other ionosondes at a similar speed.

**Supplementary Materials:** The following supporting information can be downloaded at: https://www.mdpi.com/article/10.3390/rs14030547/s1, Video S1: AM-Large_antenna, Video S2: AM-CADI_antenna, Video S3: PSK_CADI_antenna.

**Author Contributions:** Conceptualization, A.V.S. and S.P.M.; methodology, A.V.S., S.P.M.; software, A.V.S., S.P.M., K.K.G., V.A.P. and V.R.K.; set up and conducted the experiments, A.V.S., S.P.M., K.K.G. and V.A.P., data processing, A.V.S. and V.A.P.; theoretical analysis, F.I.V.; writing—original draft preparation, A.V.S.; writing—review and editing, S.P.M. and F.I.V. All authors have read and agreed to the published version of the manuscript.

**Funding:** The work is supported by a Russian Science Foundation grants #21-72-10131 (A.V.S., S.P.M. and V.A.P., Sections 1, 2 and 5), #20-12-00197 (F.I.V., K.K.G. and V.R.K., Sections 3 and 4).

**Institutional Review Board Statement:** Not applicable.

**Informed Consent Statement:** Not applicable.

**Data Availability Statement:** SDRlab 122-16 firmwares for AM and PSK fast ionosonde modes, code for receiving part of the fast ionosonde including the GNU RADIO Out-of-Tree module source code, fast ionogram receiving video demonstrations, fast ionogram examples in npz data and image file format can be found in Shindin, Alexey. (2021). The Prototype of a Fast Vertical Ionosonde Based on Modern SDR Devices-paper dataset [Data set]. Zenodo. https://doi.org/10.5281/zenodo.5795786, accessed on 17 December 2021.

**Acknowledgments:** Fedor I. Vybornov is grateful to the project No. 0729-2020-0057 within the framework of the basic part of the State assignment of the Ministry of Science and Higher Education of the Russian Federation for the technical feasibility of using CADI stations in Vasilsursk.

**Conflicts of Interest:** The authors declare no conflict of interest.

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
