# Peer review of "The Prototype of a Fast Vertical Ionosonde Based on Modern Software-Defined Radio Devices"

_remotesensing, doi:10.3390/rs14030547_

Round 1

Reviewer 1 Report

This work is a major improvement in the understanding of variations in the ionosphere. This prototype of an ionospheric sounder allows much better and accurate results of ionospheric variation. Those results are obtained with cheaper hardware and with the possibility of made variations to the SDR.

1 - I miss in the introduction more information about the application of sounding the ionosphere and where can be applied the results of this prototype.

2 - Te point 2. Materials and Methods, needs to be restructured in more subsections to be more compressive to the reader.

3 - About the line 118, all the technical parameters selected needs to be explained. For example, why you have choice 25khz and 200Hz and not another? All the technical issues needs to be more accurate explained.

4 - The different antenna used to generate all the results needs to be described more accurate and describe the main differences between all the comparatives made the best choice. The point 3. Test results it does not provide sufficient information to demonstrate the best antenna selection.

5 - The test bench designed must be described.

6 - The cost of this prototype vs the actual ionosonde needs to be compared.

7 - The abstract states that similar results are obtained with this prototype as with the CADI ionosonde, although this is not discussed in sufficient depth in the paper.

Reviewer 2 Report

Review report

  1. title: The prototype of a fast vertical ionosonde based on modern SDR devices

Authors: Alexey V. Shindin, Sergey P. Moiseev, Fedor I. Vybornov, Kseniya K. Grechneva, Viktoriya A. Pavlova, Vladimir R. Khashev

In this manuscript, a fast mode of vertical ionosonde has been introduced in detail based on the modern SDR devices at a very low price. The related modules are almost publicly available. Then, the layout of it has been clearly presented, as well as the corresponding pictures. After that, the required software to control the ionosonde and process the data near real time has been developed. Based on the constructed ionosonde in fast mode, several series of experiments have been carried out in order to valid the new instrument. The observations have been compared with an established product of CADI, which is widely employed. In my personal view, it is an interesting and useful study to introduce a new ionosonde in fast mode with a very low price. As expected, it provides a chance in future to implement a great number of ionosondes over a large area, as same as the widely distributed ground-based GNSS receivers. This manuscript is well-written. However, the authors are encouraged to consider the following major and minor concerns at first:

Major concerns:

  1. In this study, the progress to developing a new instrument has been introduced in detail. However, if you want to establish the newly ionosonde in practice, it is very important to perform the validly experiments. Then, the necessary comparisons with already employed ionosonde are essentially required. Otherwise, the observation measurements from a new instrument are doubtfully. So, the authors are strongly suggested to show more ionograms from a time series (likely in a cadence of 1 second) and then address them in detail before the comparison. Furthermore, much more examples are also urgently required to compare with the CADI measurements.
  2. As mentioned above, Figures 6 and 7 should be emphasized in detail especially in comparison with the observations from CADI. For example, the difference between these two measurements should be clarified in detail and then analysis the likely cause. Before that, can you please show the two ionograms within a closer time? Otherwise, it is not easily to compare them? Are they located at the same point to work? Much information should be addressed in detail.
  3. Here, only the ionogram from newly-developed ionsonde has been introduced. Is it possibly to detect the drift of ionosphere? Please mention it.
  4. Furthermore, the detailed comparisons with another more widely used ionsonde of DPS-4D is also needed in order to establish it solidly. With that, the specifications of advantages and shortages of the fast mode ionosonde can be convinced.
  5. I cannot play any of the videos you uploaded by using the windows mediaplayer or films and TV or another one (often employed to play .mp4 successfully). I have downloaded these videos twice from the reviewer website in forms of a compressed package or individual directly. Please check it.

Please refer to the manuscript for several minor concerns.

The end.

Reviewer 3 Report

Comments to the manuscript remotesensing-1543409

The Prototype of a Fast Vertical Ionosonde Based on Modern SDR Devices

by Alexey V. Shindin, Sergey P. Moiseev, Fedor I. Vybornov , Kseniya K. Grechneva , Viktoriya A. Pavlova, Vladimir R. Khashev

The paper attempts to solve the problem of reducing the sounding time. On the other hand, even the accumulation of data with 5-15 min resolution led to the closure of the Spidr system due to lack of data storage resources. Nevertheless, this experience must be recognized as positive. Authors should consider the following comments.

Comments

  1. Regarding the title of the paper: maybe not in the sense of a change, but it is not recommended to use abbreviations that are not widely known.
  2. Line 35, FPGA: you need to decode this abbreviation. It is advisable to give the nomenclature of the main abbreviations at the beginning or at the end of the paper.
  3. Line 59: add height resolution.
  4. Line 68: Why is the minimum frequency of 1.8 MHz chosen so high? In this case, large requirements are required on the trace of the x-component in order to obtain information on the underlying ionization from the traces of the two components.
  5. Line 106: Why is the cutoff frequency so low chosen? In what latitude range can this ionosonde be used, or is it supposed to be used for your heating experiments?
  6. Section 3. Problems of vertical sounding are associated not only with the limitation of the measurement period (in the DIDBase system, data are provided with 5 min resolution), but also with processing methods. As results of the paper I.V. Krasheninnikov, L.N. Leshchenko Errors in Estimating of the F 2-Layer Peak Parameters in Automatic Systems for Processing the Ionograms in the Vertical Radio Sounding of the Ionosphere under Low Solar Activity Conditions. Geomagnetism and Aeronomy, 2021, Vol. 61, No. 5, pp. 703–712. DOI: 10.1134/S0016793221050078 (and references in it) show, different processing algorithms can lead to very large differences in parameter values. Certain information must be given at least on the methods used to process the data (what scaling and inversion method was used?). This methodology can be crucial to the interpretation of your results. It can be assumed that there will be non-negligible errors in hmF2 data.
  7. It is necessary to give more examples of ionosonde operation.

Round 2

Reviewer 1 Report

Thank you for your response to the comments.

Author Response

Thank you very much for the review, it helped to make the paper much better!

Reviewer 2 Report

Please improve the appearance qualities of Figures 4-8 in particular in Figures 6-8 to clear the contents (such as words and numbers). Otherwise, it is not easy to read it in detail.

Author Response

(The authors gave the same response as above.)

Reviewer 3 Report

Comments to the revised manuscript remotesensing-1543409

The Prototype of a Fast Vertical Ionosonde Based on Modern SDR Devices

by Alexey V. Shindin, Sergey P. Moiseev, Fedor I. Vybornov , Kseniya K. Grechneva , Viktoriya A. Pavlova, Vladimir R. Khashev

Formally, the authors responded to all comments. I leave the decision to publish this paper at the discretion of the editor.

Author Response

(The authors gave the same response as above.)
